# Improved Mechanical, Thermal, and Hydrophobic Properties of PLA Modified with Alkoxysilanes by Reactive Extrusion Process

**DOI:** 10.3390/polym13152475

**Published:** 2021-07-27

**Authors:** Elena Torres, Aide Gaona, Nadia García-Bosch, Miguel Muñoz, Vicent Fombuena, Rosana Moriana, Ana Vallés-Lluch

**Affiliations:** 1Textile Industry Research Association (AITEX), Plaza Emilio Sala 1, 03801 Alcoy, Spain; etorres@aitex.es (E.T.); AGaona@aitex.es (A.G.); ngarcia@aitex.es (N.G.-B.); mmunoz@aitex.es (M.M.); 2Technological Institute of Materials (ITM), Universitat Politècnica de València (UPV), Plaza Ferrándiz y Carbonell 1, 03801 Alcoy, Spain; 3Division of Bioeconomy and Health, RISE Research Institutes of Sweden, Drottning Kristinas väg 61, 11428 Stockholm, Sweden; rosana.moriana.torro@ri.se; 4Centre for Biomaterials and Tissue Engineering, Universitat Politècnica de València (UPV), Camí de Vera s/n, 46022 Valencia, Spain; avalles@ter.upv.es

**Keywords:** PLA, biopolymers, alkoxysilanes, hydrophobicity, grafting, monofilament melt extrusion, thermal properties, reactive extrusion

## Abstract

An eco-friendly strategy for the modification of polylactic acid (PLA) surface properties, using a solvent-free process, is reported. Reactive extrusion (REX) allowed the formation of new covalent bonds between functional molecules and the PLA polymeric matrix, enhancing its mechanical properties and modifying surface hydrophobicity. To this end, the PLA backbone was modified using two alkoxysilanes, phenyltriethoxysilane and N-octyltriethoxysilane. The reactive extrusion process was carried out under mild conditions, using melting temperatures between 150 and 180 °C, 300 rpm as screw speed, and a feeding rate of 3 kg·h^−1^. To complete the study, flat tapes of neat and functionalized PLA were obtained through monofilament melt extrusion to quantify the enhancement of mechanical properties and hydrophobicity. The results verified that PLA modified with 3 wt% of N-octyltriethoxysilane improves mechanical and thermal properties, reaching Young’s modulus values of 4.8 GPa, and PLA hydrophobic behavior, with values of water contact angle shifting from 68.6° to 82.2°.

## 1. Introduction

The negative impact of plastics on human health and the environment is forcing the European Union to review the policy to reduce plastic waste. In 2018, European plastic production reached 61.8 million tonnes, being the global plastic production of 359 million tonnes [1]. Only 29.1 million tonnes of plastic waste were collected in the EU28+NO/CH to be treated. From 2006 to 2018, the amount of recycled plastic waste was doubled. However, 25% of plastic waste was still sent to landfills or in nature [2], from which a significant part was turned into marine litter, harming the environment. The use of biopolymers minimizes the adverse impact caused by conventional plastics because they are polymers able to be degraded to organic matter, such as water, carbon dioxide, and biomass once they up in the environment, which is a fact that decreases the accumulation of waste in the current landfills [3]. Therefore, the use of biodegradable polymers, with similar applications to conventional plastics, would be a desirable alternative to fossil-based polymers since the related environmental issues would be heavily minimized [4].

Currently, the production of biopolymers stands for about 1% of the total annual plastic production [5,6]. Nevertheless, due to social concern and the need to minimize dependence on fossil fuels, the biopolymers market is continuously growing with the estimated manufacture of 2.43 million tonnes for 2024 [7]. Polylactic acid (PLA) is one of the most demanded biopolymers, representing 13.9% of the global production of bioplastics [8], owing to its versatile characteristics [9], which make it applicable for packaging, textile, construction, and automotive applications. Specifically, in the textile industry, PLA is rapidly gaining importance due to its excellent mechanical properties when the polymer is processed into yarns [10,11], being comparable to the commercial counterparts such as polyethylene (PE), polystyrene (PS), and polyethylene terephthalate (PET) [12,13]. Nonetheless, although PLA textiles exhibit excellent hand, moisture management, and easy-care properties [14], PLA fibers, unfortunately, are susceptible to degrade in wet conditions, which can be induced by the hydrolysis of ester bonds, leading to the decrease of the degree of polymerization [15]. Substantial research has suggested that PLA textiles undergo degradation during home laundering and storage; in such a way, there has been observed a 20% and 25% decrease in filament strength and modulus, respectively, after 10 times washing at pH 10, 55 °C [16]. Furthermore, a 50% decrease of the polymer MW was observed after 262 days of storage at 40 °C and 80% humidity [17]. Therefore, improving the hydrophobicity character of PLA fibers will enhance PLA resistance to hydrolysis, and hence, ameliorate its durability [18].

Additionally, in home textile applications, self-cleaning or easy-cleaning functionalities are usually required, which can be achieved by increasing the polymer surface hydrophobicity [19]. For instance, new materials with high hydrophobicity and durability properties have been developed through the formation of micro-thin super-slick hydrophobic surfaces preventing the adhesion of ice, dirt, debris, mold, snow, etc. [20]. In addition, within the textile market, self-cleaning added value functionalities are currently gaining importance. In order to achieve such functionalities, the most extended strategy is the application of coating or surface modification techniques [21]; however, the textile product loses its efficiency with the usages. Polymer functionalization with additives by compounding technology is another approach to obtain added value textiles. This process is based on a physical mixture of polymer and additive in the molten state [22]. Nevertheless, in many cases, the final product eventually loses the added functionality, since the additives migrate to the surface and are released to the environment [23]. Recently, reactive extrusion (REX) is being used to overcome the aforementioned drawbacks, considering that functional molecules are anchored into the polymeric matrix. REX technology provides stable and irreversible covalent bonds, achieving a uniform distribution of functional moieties into the polymer matrix and avoiding particle aggregation, migration, and leaching [24,25,26]. Furthermore, reactive extrusion offers a one-step solvent-free route to produce novel and high-performance materials with new functionalities [27]. Therefore, the REX process allows the incorporation of additives into the polymer matrix through covalent bonds under milder and environment-friendly conditions.

Accordingly, to improve PLA textile performance, new functionalities such as self-cleaning, dust removal, and durability can be granted to the biopolymer by the modification of PLA surface hydrophobicity [19]. Considering that this characteristic strongly depends on the surface free energy [28], it can be tuned by chemical modifications via reactive extrusion. Indeed, the incorporation of alkoxysilanes into polymeric matrices is proposed to modify their surface free energy and, consequently, their hydrophobicity [29,30,31]. Therefore, the use of biodegradable PLA and reactive extrusion as a processing method involves a cost-effective and environmentally friendly approach thanks to the extruder’s unique stability as a vehicle for carrying out chemical reactions in the bulk phase; besides, it is a continuous process with low reaction times compared to batch processing and free-solvent, which contributes to minimizing the environmental footprint [32].

In the present study, PLA modifications through the grafting of alkoxy derivatives, using a single-step process (REX), are reported for conferring hydrophobicity to this biopolymer. Thereupon, the PLA matrix was covalently modified with two different alkylalkoxysilanes (R’-Si-OR). The covalent incorporations were confirmed by ^1^H NMR spectroscopy. Hydrophobic, thermal, and mechanical properties were studied after the obtention of flat tape samples through a conventional monofilament extrusion process.

## 2. Materials and Methods

### 2.1. Materials and Reagents

Polylactic acid (PLA) Ingeo^TM^ 6201D with a number average molecular weight of 97,000 g·mol^−1^ (M_n_) was provided by NatureWorks LLC (Minnetonka, MN, USA), which has a range melting temperature from 155 to 170 °C. In order to develop hydrophobic PLA, different alkoxy derivatives were employed. Specifically, phenyltriethoxysilane, Ph-Si(OEt)_3_, with a minimum purity of 98%, was supplied by Sigma-Aldrich (Madrid, Spain). A second organosilane, N-octyltriethoxysilane, Oct-Si(OEt)_3_, with a purity of 97% was supplied by abcr GmbH (Karlsruhe, Germany).

### 2.2. Sample Preparation

#### 2.2.1. Reactive Extrusion

In the reactive melt processing, a screw speed of 300 rpm, feed rate of 3 kg·h^−1^, and profile temperature between 150 and 180 °C were set as optimal extrusion conditions. According to previous experience and knowing that around 0.3 wt% of alkylalkoxysilane corresponds to the theoretical stoichiometry, which considers that each PLA hydroxyl (-OH) end-group reacts with one alkoxy end-group from the alkoxysilanes (1:1), the exchange reaction yield obtained was very low. Accordingly, the alkoxysilane molar concentration was estimated considering that, with less than 50% probability, only one alkoxy end-group from alkoxysilanes reacts with one PLA hydroxyl (-OH) end-group (3:1), corresponding to 1.3–1.5 wt% depending on the organosilane molar mass (Table 1). Thus, 1.3 and 3.0 wt% were established as suitable mass concentrations. On the other hand, having the limitation of a minimum flow rate of 0.4 mL·min^−1^ provided by the HPLC pump, it was not possible to perform the reaction process with mass concentrations lower than 0.8 wt% in order to guarantee the residence time required. Such alkoxysilanes were introduced without solvent dilution using an HPLC pump. Thus, Table 1 summarizes the functionalized PLA samples obtained by the reactive extrusion approach. In a typical experiment, neat PLA pellets were dried in a vacuum oven overnight at 75 °C and then extruded with different percentages of alkoxysilanes, according to Table 1, in a twin-screw extruder (Leistritz ZSE 18, Nuremberg, Germany) with L/D = 68 (L and D = 18 mm are the length and diameter of the screw, respectively). The material was subsequently cooled in a water bath and pelletized. The obtained pellets were dried in a vacuum oven overnight at 75 °C in order to remove the remaining water.

#### 2.2.2. Flat Tape Extrusion

New functionalized PLA materials, developed by REX, were subsequently processed in a monofilament extrusion pilot plant to obtain flat tape samples. This pilot plant is composed of four basic elements/stages: single-screw extruder, spinning die, drawing unit composed by godets, ovens and cooling baths, and a winder. More specifically, the extrusion unit is formed by a melt pump, which works at flow rates inferior to 3 kg·h^−1^, and a single-screw extruder with a diameter of 25 mm and L/D = 24:1. Finally, flat tapes were prepared using the conditions specified in Table 2.

#### 2.2.3. Characterization Methods

Total silicon contents in flat tape samples were obtained by inductively coupled plasma optical emission spectroscopy (ICP-OES), using an Agilent 5110 equipment (Santa Clara, CA, USA). The average values and standard deviations were calculated measuring five replicates per sample. The total integrity and covalent incorporation of functional alkoxysilanes were determined by liquid-state ^1^H NMR spectroscopy. High-resolution liquid-state ^1^H NMR spectroscopy measurements were carried out with a Bruker AV400 instrument (Mannheim, Germany) working at 300 MHz. The analyses were performed using approximately 20 mg of sample and dissolved in 0.6 mL of deuterated chloroform (Sigma-Aldrich, Madrid, Spain). Chemical shifts were quoted in ppm and referred to internal tetramethylsilane (TMS).

Neat and functionalized PLA were characterized by size exclusion chromatography by a high-performance liquid chromatography photodiode array detection method using a uHPLC 1260 Infinity Binary LC System (Santa Clara, Agilent). In a typical analysis, samples were dissolved in chloroform as elution solvent, with a 2 wt% concentration and filtered through 0.22 μm pore size PTFE filters prior to measurements. Number-average (M_n_) weight-average (M_w_) molar masses and dispersities (D) were obtained from a calibration curve based on polystyrene (PS) standards. Empower 3 was used for processing data.

Differential scanning calorimetry (DSC) analyses were carried out (50 mL·min^−1^ of nitrogen atmosphere, 99.99% purity) in a DSC 3+/Mettler Toledo (Barcelona, Spain). Measurements to determine the melting and crystallization behavior of neat and functionalized PLA were performed by encapsulating 10–12 mg of sample in an aluminum pan and following this temperature program: (i) heating from 30 to 170 °C, corresponding to the holding temperature, T_hold_, (ii) isothermal step of 2 min at T_hold_ to remove thermal history, (iii) cooling ramp from T_hold_ to −30 °C and, (iv) heating ramp from −30 to 250 °C. All tests were performed at a heating/cooling rate of 20 °C·min^−1^. As the previous thermal history was removed, the crystallization temperature (T_c_) was established from cooling scans, whereas the glass transition temperature (T_g_) and melting temperatures (T_m_) were determined using the information provided by the second heating scan. The degree of crystallinity (X_c_) was calculated in the first heating program by means of Equation (1), where ∆H_m_ is the melting enthalpy of PLA (J·g^−1^), ∆H_mc_ corresponds to the cold crystallization enthalpy of PLA (J·g^−1^), ∆H^0^_m_ is the melting enthalpy associated with a theoretically fully crystalline PLA, reported to be 93 J·g^−1^ [33,34], and *x* is the the proportion of pure PLA in the different samples with additives:(1)$$X_{c}\left(\%\right)=\left[\frac{\Delta H_{m\;}-\Delta H_{\mathrm{mc}}}{\Delta H_{m}^{0}\cdot x}\right]\cdot 100$$

Thermogravimetric analyses (TGA) were conducted under nitrogen atmosphere (50 mL·min^−1^) with a Mettler Toledo TGA/SDTA 851E analyzer (Barcelona, Spain). In all cases, 8 mg of sample were heated at 10 °C·min^−1^ from 25 to 700 °C.

Mechanical characterization of flat tapes was performed by tensile tests. Mechanical trials were carried out at room temperature with an INSTRON 3343-K7523 (Barcelona, Spain) machine following UNE EN ISO 13934-1. Samples were tested using 0.5 kN load cell a crosshead speed of 50 mm·min^−1^ and a distance between clamps of 50 mm. Typically, to achieve an accurate average value of tensile strength, Young’s modulus, and elongation at break, at least five repetitions of each sample were performed following UNE EN ISO 13934-1 recommendations.

Changes in the wetting properties of functionalized PLA were determined by water contact angle measurements (WCA) on the sample’s dry surfaces by the sessile drop technique. Neat PLA was used as control. An EasyDrop Standard goniometer, model FM140 supplied by KRÜSS (KRÜSS GmbH, Hamburg, Deutschland), with a measurement range from 1° to 180° and precision of ± 0.1° was used for this purpose. The equipment was supplemented with a video capture kit and analysis software (Drop Shape Analysis SW21; DSA1). Five replicates of each sample were tested, performing 10 measurements for each replicate, and calculating the mean value from 50 values per sample with a standard deviation lower than 3%.

## 3. Results

### 3.1. PLA/Alkoxysilanes Reaction

The reaction between R’-alkoxysilanes (-R’ = phenyl or octyl) and PLA through hydroxyl (-OH) end groups was expected to lead to the formation of R’-Si-PLA by exchange reactions, as shown in Figure 1 [23,35].

After chemical modification of neat PLA, ICP-OES analyses were performed to determine total silicon contents into the PLA matrix (grafted and non-grafted). More specifically, Table 3 shows concentration (ppm) and weight percentage of silicon into a functionalized PLA matrix, as well as the incorporation ratio of organosilane units (%), which are calculated from the ratio between experimental and theoretical silicon contents.

Table 3 shows the presence of alkoxysilanes in the PLA polymeric matrix. Silicon weight percentage on the final samples was much higher for PLA modified with phenyltriethoxysilane (P-PLA-A and P-PLA-B) than PLA modified with octyltriethoxysilane (O-PLA-A and O-PLA-B). Therefore, it can be presumed that the reaction is affected by steric factors [36], where pendant octyl groups are subjected to higher steric hindrance, hampering the exchange reaction between PLA hydroxyl end-group and the organosilane. In this sense, the length of Si-octyl (8.54 Å) and Si-phenyl (4.24 Å) was predicted through ChemDraw software to support such an assumption.

Notwithstanding, ICP analyses only confirmed the presence of silicon species into the polymeric matrix but did not provide accurate information about new covalent bonds formed between the PLA backbone and alkoxy derivatives. To this end, the efficient anchoring of these alkoxy derivatives, by a reactive extrusion process, was verified by the liquid ^1^H NMR technique. Figure 2 shows the reaction scheme between PLA and N-octyltriethoxysilane (Figure 2a), as well as the ^1^H NMR spectra for functionalized (O-PLA-A and O-PLA-B) and neat PLA samples (Figure 2b). According to the literature [37,38], the PLA spectrum revealed four main signals at 1.6, 5.2, 1.4, and 4.3 ppm, which correspond to CH_3_ (a), CH (b), CH_3_ (c + r), and CH (d) next to the terminal groups, respectively [39,40]. However, when PLA was modified with Oct-Si(OEt)_3_, new signals associated to N-octyltriethoxysilane were detected. In this sense, signals at 1.2, 0.9, and 0.6 ppm were observed when alkoxysilane was incorporated. More specifically, chemical shifts at 1.2 were assigned to protons p and f-k from unreacted ethoxy groups (CH_3_, p) and the octyl chain (f–k). On the other hand, signals at 0.9 and 0.6 ppm were assigned to the terminal CH_3_ (l) and the methylene was linked to silicon CH_2_ (e) (Figure 2a). Moreover, a new quadruplet appeared at 3.8 ppm (Figure 2b), being assigned to the methylene group (CH_2_O) of unreacted alkoxysilane ethoxy groups (m, Figure 2a). This result was expected, as the ethoxy groups were introduced in large excess compared to the terminal hydroxyl functions of PLA. Therefore, these chemical signals confirmed the presence of alkoxysilane into a PLA matrix, whereas the formation of new covalent bonds through PLA hydroxyl end groups was evidenced with the appearance of a chemical shift at 3.7 ppm (d’), which was assigned to the resonance modification of the terminal methine group in the α-position of the hydroxyl function (Figure 2a). Thus, signal d shifted, from 4.2 (d) to 3.7 ppm (d’), due to the modification of a methine group chemical environment. Such a chemical signal is shifted compared to the value observed from starting PLA, which indicates that the reagent is no more in its initial form once the alkoxysilane is covalently incorporated (see inset, Figure 2). Therefore, the presence of this new quadruplet (3.7 ppm, d’) and the complete disappearance of a d signal (4.2 ppm) confirmed the formation of covalent bonds between the PLA matrix and N-octyltriethoxysilane by exchange reactions.

On the other hand, the integration of proton signals from the modified methine group in the α-position of the hydroxyl function (d’) and signal of the unreacted ethoxy group (p) was used to estimate the evolution of grafting reaction, corresponding to the d’/p ratio (Table 4). By integrating chemical shifts, we found a clear limitation in the reaction yield between alkoxy units (-OR) and PLA hydroxyl (-OH) end groups, as well as an excess of reactive ethoxy end groups, as the d’/p ratio decreased when the octylalkoxysilane concentration was increased. Additionally, the doublet at 1.4 ppm and 1.5 ppm assigned to CH_3_ (c) next to the terminal -OH group and CH_3_ (r) next to the terminal -COOH group, respectively, disappeared when 1.5 wt% of N-octyltriethoxysilane was added to the PLA matrix, implying that the reaction proceeds through both terminal groups. However, when the percentage of N-octyltriethoxysilane was increased up to 3.0 wt% (O-PLA-B), the doublet assigned to CH_3_ (r) next to the terminal -COOH group remains visible, suggesting that the reaction proceeds mainly through PLA hydroxyl (-OH) end groups [35]. However, other side reactions cannot be excluded, such as transesterifications reactions through the PLA carboxylic acid (-COOH) end groups or in situ condensation of the organosilane. 

Comparable effects were observed when PLA was modified with phenyltriethoxysilane (Figure 3). In the case of P-PLA-A and P-PLA-B samples, chemical shifts assigned to phenyl organic fragments were revealed between 7 and 7.5 ppm (e–i). Likewise, the spectra depicted a new proton signal at 3.7 ppm (d’), confirming the resonance modification of the terminal methine group in the α-position of the hydroxyl function. Similar to O-PLA samples, the total disappearance of the signal related to methine next to the terminal group (d, 4.2 ppm) verified the phenyltriethoxysilane incorporation by exchange reactions. On the other hand, new signals at 1.3 (p) and 3.8 ppm (m), associated with the unreacted ethoxy groups, were detected and integrated. Accordingly, the d’/p ratio suggested a higher reactivity of terminal ethoxy groups from phenyltriethoxysilane than octyltriethoxysilane, as the signal intensity associated with unreacted ethoxy groups (p) was considerably lower for P-PLA-samples than O-PLA-samples (Table 4). Inferior intensities for the p signal are associated with higher reaction yield, whether due to exchange reactions or organosilane hydrolysis/self-condensation in situ during the process. The high incorporation observed by ICP analysis and the remaining presence of signals at 1.4 and 1.5 ppm, assigned to CH_3_ (c) next to the terminal -OH group and CH_3_ (r) next to the terminal -COOH group, respectively, suggested that the hydrolysis and self-condensation of phenyltriethoxysilane molecules also occurred in situ during the reactive extrusion process, leading to the formation of highly condensed silicon species (inset Figure 3). Therefore, the reactive extrusion process allowed both alkoxysilane hydrolysis-condensation and exchange reactions through terminal hydroxyl and ester functions. However, the doublet assigned to CH_3_ (r) next to the terminal -COOH group remains visible for P-PLA samples, suggesting that the exchange reaction proceeds mainly through PLA hydroxyl (-OH) end groups. More information about the reaction scheme of the PLA modification with (3-glycidyloxypropyl)trimethoxysilane, phenyltriethoxysilane and titanium phenoxide can be found in the Appendix A. 

Aiming at verifying such hypotheses, a deeper investigation on the effect of alkylalkoxysilane incorporation on PLA molar mass was performed by size exclusion chromatography (SEC) experiments. SEC results for neat and functionalized PLA are shown in Table 5. As reported in the literature, PLA is susceptible to thermal degradation during the melt extrusion process, leading to a decrease in M_w_, rheological, and mechanical properties [41,42]. Accordingly, after PLA extrusion processing, the results showed a decrease in M_n_ (from 97,000 to 75,000 g·mol^−1^) and M_w_ (from 173,000 to 116,000 g·mol^−1^), suggesting the molecular degradation of PLA. In the same way, SEC results for functionalized PLA with Octyl-Si(OEt)_3_ evidenced the highest reduction in molar mass values, which may be attributed to chain scissions induced by ester-alkoxysilane exchange reactions at high temperatures [23,38]. In the case of phenyltriethoxysilane, only a slight reduction in SEC values was observed despite the higher presence of alkoxysilane species, such as those previously evidenced by ICP (Table 3).

### 3.2. Characterization of the Functionalized Polymers

Thermal properties of functionalized PLA flat tape samples were studied taking neat PLA as a reference. Glass transition and melting endotherms registered during the second heating scan are shown in Figure 4. All curves exhibited small glass transition temperature peaks (T_g_) and a broad melting endotherm signal (T_m_). More specifically, neat PLA flat tape displayed two small glass transition temperature peaks, which are directly related to polymeric chain movement, at T_g1_ = 71 °C and T_g2_ = 82 °C (Figure 4, Table 6). The presence of two glass transition peaks may be associated with the formation of two regions with different crystallinity levels as a result of the drawing process. After grafting 1.5 wt% of octyltriethoxysilane and 1.3 wt% of phenyltriethoxysilane, both peaks were slightly shifted toward higher temperatures. Interestingly, when 3.0 wt% of octyltriethoxysilane and 2.7 wt% of phenyltriethoxysilane were incorporated through a reactive extrusion process, only one glass transition peak was observed at lower temperatures (<70 °C). This thermal effect on functionalized flat tapes suggested a lubricant effect at high concentrations of alkoxysilanes, which enabled a greater segmental motion and chain sliding, requiring thus lower stress forces for both processes [43]. Regarding melting temperature, the second heating scan provided similar melting curves for PLA and functionalized samples, where a wide melting peak around 170 °C was obtained. However, samples with low alkoxysilane concentrations, O-PLA-A and P-PLA-A, displayed a small shoulder overlapped to the melting peak (highlighted in the Figure 4 with a circumference), which is associated with the fusion of imperfect grown crystals, whereas large melting peaks are related to highly ordered crystalline structures formed after drawing stages [23,44,45]. Therefore, flat tape samples with low alkoxysilane concentrations showed two regions with different crystallinity, while high contents provided a homogeneous crystal region, probably as a result of the appearance of nucleating points, which induced a more uniform crystal growth [41], and a lubricant effect that enabled a higher and homogeneous drawing treatment.

Table 6 lists several characteristic parameters established from DSC thermograms, such as glass transition temperature (T_g_), melting temperature (T_m_), melting enthalpy (∆H_m_), and degree of crystallinity (X_c_) obtained from the first heating program. Polymer crystallinity refers to the alignment of the polymer molecular chains and can affect optical, mechanical, and thermal properties after processing upon cooling from melting temperature and applying mechanical stretching. Therefore, after calculating the crystallinity degree of PLA flat tapes (Table 6), there can be discerned an increase in crystallinity after reaction with the alkoxysilanes, being enhanced with the incorporation of higher contents. Therefore, the incorporation of alkoxysilanes into the PLA matrix increases crystallization probably because of the appearance of nucleation points [46]. Moreover, as reported in Table 2, alkoxysilane incorporations into the PLA matrix allowed higher draw ratios for flat tape processing, suggesting a lubricant effect that enabled a greater segmental motion. Such a rheological effect provided higher longitudinal alignment through stretching forces and, consequently, an increase in the crystallinity [47,48,49]. Therefore, PLA samples with a higher incorporation of functional moieties reached higher crystallinity degrees, possibly owing to both nucleation effect at the coupling sites and higher draw ratio appliance, increasing from 47% for pure PLA to values close to 56% for samples with the highest concentrations of alkoxy derivatives.

On the other hand, the relationship between the degree of disintegration under composting conditions in the different samples of neat PLA and modified PLA can be estimated through their crystallinity. The previous literature has shown in multiple studies that an increase in crystallinity can lead to a delay in the biodegradation processes [50,51,52]. These biological processes are normally carried out by lipases, proteases, and esterases secreted from microorganisms in soil compost, and they have a greater facility to act in amorphous domains. However, multiple studies show that despite considerably increasing the crystallinity of a PLA, the delay in the disintegration processes is only a few days. For example, Balart et al. [53] reported a crystallinity in modified PLA with the addition of lignocellulosic fillers and bio plasticizers of twice the initial value. However, the degree of disintegration was delayed by only 14 days. In another work, Dominguez-Candela et al. [54] reported an increase of more than 53% in PLA crystallinity when introducing an organic bio plasticizer, but, a disintegration of more than 90%, which is considered by the UNE 20,200 standard as a biodegradable material, was obtained only 3 days later. Therefore, in view of the previous results, and given that in the present study, there is only a 10.6% increase in crystallinity concerning neat PLA, it can be affirmed that a subsequent disintegration process would be practically unaffected, considering the PLA modified by REX as totally biodegradable.

Figure 5 displays thermogravimetric analysis (TGA) for neat and functionalized PLA samples. In general, the results confirmed that PLA functionalized with different alkoxysilane contents presented similar degradation temperatures to neat PLA, which agrees with previous studies [40,42]. However, when TGA results were analyzed in detail (inset Figure 5 and Table 7), slight variations in thermal stabilities were observed. More specifically, PLA samples modified with low alkoxysilane concentrations, O-PLA-A and P-PLA-A, displayed similar thermal stability to neat PLA, breaking down at temperatures around 364–362 °C, whereas higher organosilane contents showed slightly lower decomposition temperatures (356–358 °C). As evidenced by 1H-NMR and SEC analyses, these small changes may be associated with ester-alkoxysilane exchange reactions, which occurs at high temperature, causing chain scissions via C-H hydrogen transfer reactions [36,54,55,56,57]. Consequently, lower molar weights and, therefore, lower thermal stabilities were obtained compared to neat PLA.

The mechanical properties of neat and functionalized PLA flat tape samples were evaluated using an Instron mechanical testing machine. As shown in Table 8, results confirmed that the covalent incorporation of functional moieties into the PLA matrix did not reduce the inherent PLA mechanical properties. More specifically, Young’s modulus, tensile strength at break, elongation at break, and dimensions of the flat tape samples were gathered in Table 8.

As listed in Table 8, results confirmed the non-detrimental influence on mechanical properties when functional moieties were grafted into the PLA matrix. In this sense, functionalized samples depicted slightly superior tensile strength and Young’s modulus, as well as lower elongation at break than neat PLA, which is related to the application of higher draw ratios and, consequently, superior longitudinal alignment of polymer molecular chains [58]. According to DSC results, the presence of alkoxysilanes into the PLA backbone caused a lubricant effect that allowed a more severe drawing treatment, reaching greater ordered crystalline structures and, consequently, increasing the Young’s modulus. This lubricant effect was more pronounced when the alkoxysilane content was raised, allowing the application of draw ratios up to 6.2 and 6.6 for O-PLA-B and P-PLA-B, respectively [23,40,42,43]. Conversely, the highest alkoxysilane concentrations and draw ratios did not report the highest Young’s modulus. Specifically, in the case of O-PLA-B, this fact may be associated with the scission chains caused by ester-alkoxysilane exchange reactions, which slightly reduce the molar weight, thermal stability, and, finally, mechanical behavior of PLA flat tapes [42,59]. For the P-PLA-B sample, based on ICP and NMR results, higher phenyltriethoxysilane led to the formation of highly silicon condensed species acting as particle aggregates and, consequently, as defects in the polymer matrix. Therefore, functionalized PLA samples obtained using 1.3% alkoxysilane concentration, O-PLA-A, showed the best mechanical properties, reaching values of Young’s modulus and elongation at break close to 5.1 GPa and 30%, respectively.

Hydroxyl end groups afford an opportunity to transform, through chemical modification, partial wettability of PLA into a low surface energy polymer, thus yielding hydrophobicity. This approach was carried out by functionalization of the PLA backbone with different alkoxysilanes in the molten state. The static water contact angle (WCA) for neat and hydrophobized PLA flat tapes was measured to evaluate the level of hydrophobicity attained after modification.

Figure 6 shows contact angle values obtained for neat and grafted PLA using water as the test liquid, providing a contact angle average value of 68.6° for neat PLA. This average value is in concordance with a previous study reported by Jordá-Vilaplana et al. [60]. Taking into account that a contact angle over 90° is considered a hydrophobic surface, and values lower than 30° are regarded as hydrophilic surfaces [61], untreated PLA shows a slightly hydrophobic surface behavior. As shown in Figure 6, grafting PLA with low surface energy components, such as Ph-Si(OEt)_3_ and Oct-Si(OEt)_3_, minimizes the polymer surface free energy, giving place to an increase in the hydrophobicity [23]. More specifically, in agreement with its higher non-polar nature, PLA grafted with Oct-Si(OEt)_3_ showed a greater contact angle, achieving a WCA of 82.2° and therefore an increase in the hydrophobicity of almost 20% compared with neat PLA. Relating the results shown in the hydrobicity and mechanical tests, it appears that the functionalized samples, on the one hand, possess higher hydrophobicity and on the other hand superior mechanical strength properties due to the superior longitudinal alignment of polymer molecular chains, as Yan et al. demonstrated [58]. In addition, the insertion of the polar groups present in molecules such as Oct-Si(OEt)_3_ may lead to a higher interaction between PLA molecules, restricting their mobility and increasing the mechanical properties.

Aiming at verifying the stability of the grafted alkoxysilanes by the formation of new covalent bonds, samples were treated in hot water (T = 80 °C) for 24 h under stirring conditions. The results showed that after washing treatment, WCA remained almost invariable, confirming the formation of stable and irreversible covalent bonds between PLA backbone and alkoxysilanes molecules through exchange reactions.

## 4. Conclusions

PLA chemical modification by alkoxysilanes in the molten state allowed the improvement of its native hydrophobicity and mechanical properties. After the reactive extrusion process, ICP-OES analyses confirmed the presence of alkoxy derivatives into the PLA matrix. ^1^H NMR spectra confirmed the effective grafting process by exchange reactions. In addition, ^1^H NMR results agreed with size exclusion chromatography analyses (SEC), which evidenced a slight reduction in molar mass values for functionalized PLA, being associated with chain scissions induced by ester-alkoxysilane exchange reactions at high temperature.

After evaluating thermal properties, flat tapes of neat PLA and PLA grafted with low alkoxysilanes concentrations displayed two small glass transition temperature peaks, which are associated with the formation of two regions with different crystallinity levels as a result of the drawing process. However, for higher alkoxysilanes concentrations, only one glass transition peak was observed at lower temperatures, which is associated with a uniform drawing process thanks to the creation of nucleating points and a lubricant effect. Similar observations were obtained regarding melting temperatures. Finally, the crystallinity degree was enhanced with the incorporation of higher contents of alkoxysilanes into the PLA matrix, due to the appearance of nucleation points.

Mechanical properties evaluation displayed an enhancement on tensile strength and Young’s modulus in all cases, which is associated with the increase in the crystallization degree. However, the highest alkoxysilane concentrations did not report the highest values; in the case of O-PLA-B, it may be associated with the scission chains caused by ester-alkoxysilane exchange reactions, whereas for the P-PLA-B sample, it may be associated with the formation of highly silicon condensed species acting as particle aggregates and, consequently, as defects in the polymer matrix. Therefore, PLA samples obtained using 1.3% alkoxysilane concentration (O-PLA-A) showed the best mechanical properties, reaching values of Young’s modulus and elongation at break close to 5.1 GPa and 30%, respectively.

Finally, water contact angle values showed a clear variation depending on the type of alkoxysilane and concentration used, obtaining the highest hydrophobicity when 3.0 wt% of Oct-Si(OEt)^3^ was incorporated (O-PLA-B) into the polymer matrix. Furthermore, after applying a washing treatment, similar WCA values were obtained, confirming the stability of the grafted alkoxysilanes by the formation of stable and irreversible covalent bonds between PLA backbone and alkoxysilanes molecules.

## Figures and Tables

**Figure 1 polymers-13-02475-f001:**
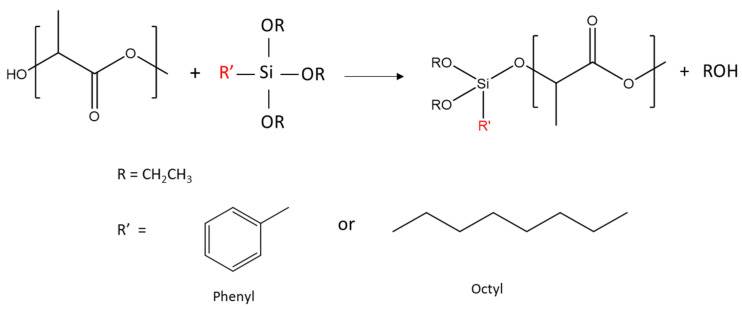
Reaction schemes between PLA hydroxyl end groups and alkoxysilanes. Only OR exchange is shown here.

**Figure 2 polymers-13-02475-f002:**
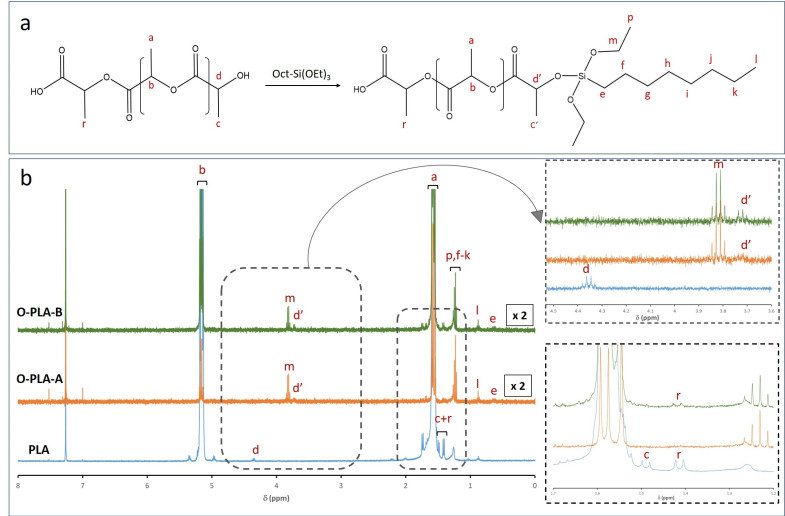
(**a**) Reaction scheme for PLA modification with N-octyltriethoxysilane and (**b**) ^1^H NMR of neat and functionalized PLA with different concentrations of N-octyltriethoxysilane: O-PLA-A (1.5 wt%) and O-PLA-B (3.0 wt%).

**Figure 3 polymers-13-02475-f003:**
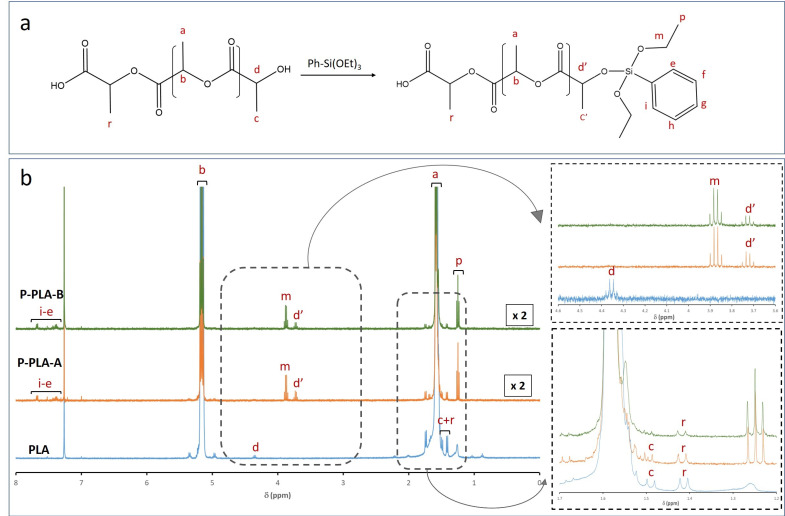
(**a**) Reaction scheme for PLA modification with phenyltriethoxysilane and (**b**) ^1^H NMR of neat and functionalized PLA with different phenyltriethoxysilane concentrations: P-PLA-A (1.3 wt%) and P-PLA-B (2.7 wt%).

**Figure 4 polymers-13-02475-f004:**
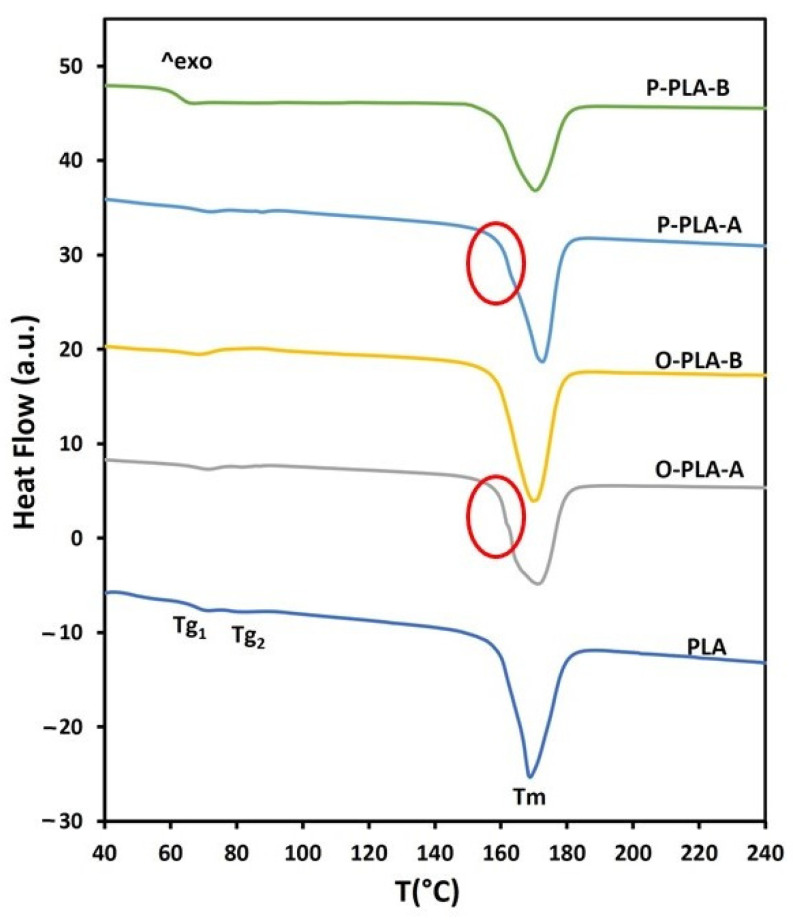
DSC thermograms for neat and functionalized PLA flat tapes obtained by reactive extrusion.

**Figure 5 polymers-13-02475-f005:**
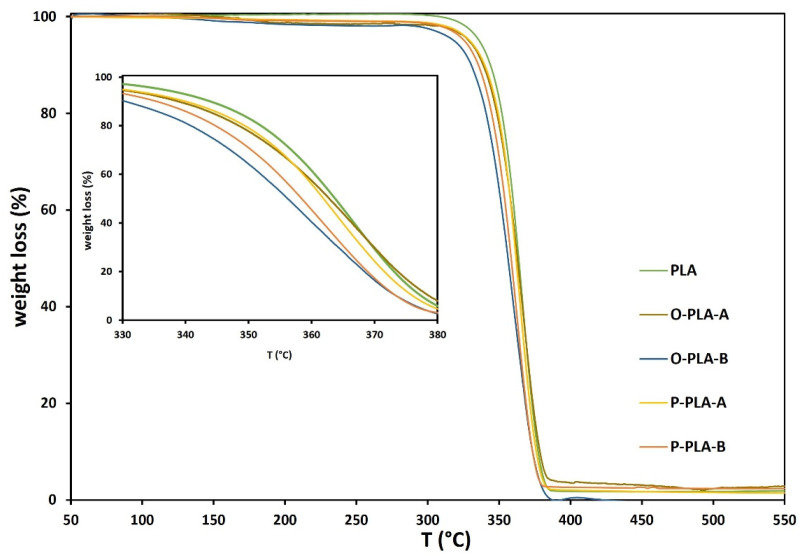
DSC thermograms for neat and functionalized PLA flat tapes obtained by reactive extrusion.

**Figure 6 polymers-13-02475-f006:**
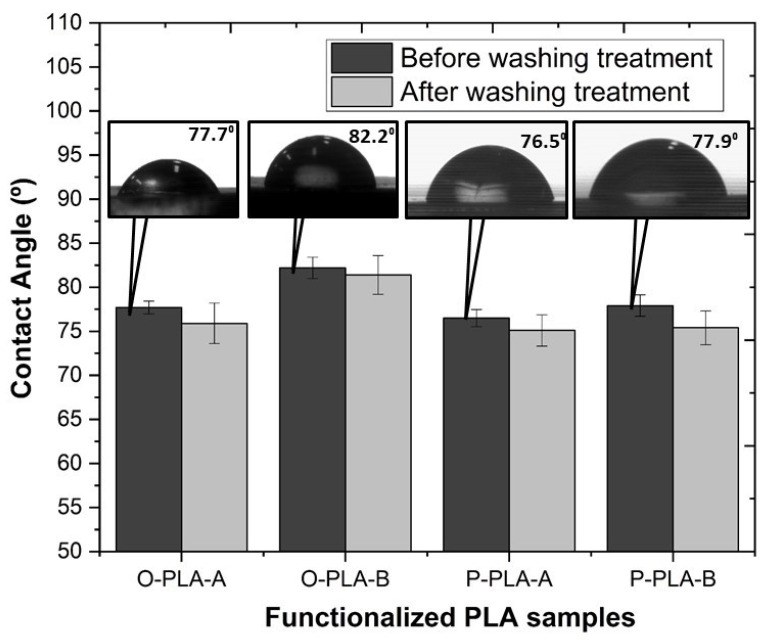
Variation of the water contact angle on the neat and functionalized PLA flat tape samples before and after washing treatment.

**Table 1 polymers-13-02475-t001:** Functionalized PLA samples through reactive extrusion.

Sample	Functional Molecule (R’)	Concentration (wt%)	Molar Concentration (mol/h)
PLA (control)	-------	-------	0.031
O-PLA-A	Oct-Si(OEt)_3_	1.5	0.167
O-PLA-B	Oct-Si(OEt)_3_	3.0	0.337
P-PLA-A	Ph-Si(OEt)_3_	1.3	0.167
P-PLA-B	Ph-Si(OEt)_3_	2.7	0.337

Reactive extrusion conditions: Temperature profile = 150–180 °C, Screw speed = 300 rpm, PLA flow rate = 3 kg·h^−1^, and alkoxysilane flow rates = 0.4–1.5 mL·min^−1^.

**Table 2 polymers-13-02475-t002:** Melt extrusion conditions for flat tape processing using monofilament melt spinning technology.

Sample	V_godet1_ (m/min)	V_godet4_ (m/min)	Draw Ratio (DR)	Dimensions (Width × Thickness, mm)
PLA	10	45	4.5	2.70 × 0.14
O-PLA-A	10	55	5.5	2.34 × 0.13
O-PLA-B	10	66	6.6	2.40 × 0.14
P-PLA-A	10	55	5.5	2.32 × 0.13
P-PLA-B	10	62	6.2	2.20 × 0.14

Flat tape extruding conditions: Temperature profile = 195/205/205/220/215 °C, Flow rate = 1 kg·h^−1^, Cooling bath temperature = 35 °C and Godet speed range = 10–66 rpm.

**Table 3 polymers-13-02475-t003:** Chemical analyses of functionalized PLA flat tapes determined by ICP-OES.

Sample	Concentration (ppm)	Si (%)	Incorporation (%) ^a^
O-PLA-A	259 ± 16	0.026	16.6
O-PLA-B	364 ± 22	0.036	12.2
P-PLA-A	727 ± 55	0.073	48.2
P-PLA-B	1594 ± 410	0.159	50.4

^a^ Incorporation percentage calculated as the ratio between the experimental and theoretical concentration of silicon atoms in the PLA matrix (wt%).

**Table 4 polymers-13-02475-t004:** Chemical shifts and integration signal values.

Sample ID	Assignment	Chemical Signal (ppm)	Integration ^a^
O-PLA-A	d’	3.7	1
	p	1.21	9.58
O-PLA-B	d’	3.7	1
	p	1.21	12.61
P-PLA-A	d’	3.7	1
	p	1.21	3.45
P-PLA-B	d’	3.7	1
	p	1.21	6.59

^a^ Values obtained from the integration of the d’ and p chemical shifts using MestreNova software.

**Table 5 polymers-13-02475-t005:** SEC analysis on neat and functionalized PLA with different alkoxysilane concentrations.

Sample	M_n_ (g·mol^−1^)	M_w_ (g·mol^−1^)	D
Neat PLA	97,000 ± 20,000	173,000 ± 20,000	1.85
Extruded PLA	75,000 ± 20,000	116,000 ± 20,000	1.56
O-PLA-A	48,000 ± 6000	94,000 ± 6000	1.96
O-PLA-B	45,000 ± 9000	90,000 ± 15,000	1.99
P-PLA-A	61,000 ± 6000	101,000 ± 4000	1.66
P-PLA-B	66,000 ± 4000	117,000 ± 2000	1.62

**Table 6 polymers-13-02475-t006:** DSC characteristic parameters of neat and functionalized PLA flat tapes by reactive extrusion.

Sample	T_g1_ (°C)	T_g2_ (°C)	T_m_ (°C)	∆H_m_ (J·g^−1^)	X_c_ (%) ^a^
PLA	71	82	168	43.8	47
O-PLA-A	72	88	170	48.4	52
O-PLA-B	67	----	169	52.6	56
P-PLA-A	72	82	171	48.1	51
P-PLA-B	69	----	170	48.4	52

^a^ Crystallinity was estimated according to the ratio between the heat of fusion (J·g^−1^) measured and tabulated value for 100% crystalline PLA (93.0 J·g^−1^).

**Table 7 polymers-13-02475-t007:** Main thermal parameters of neat and functionalized PLA flat tapes by reactive extrusion obtained by TGA analysis.

Sample	T_0_ (°C)	T_50_ (°C)	T_f_ (°C)
PLA	292	364	385
O-PLA-A	294	363	385
O-PLA-B	292	356	387
P-PLA-A	298	362	385
P-PLA-B	298	358	380

T_0_: starting degradation temperature, T_50_: temperature when 50% mass is degraded, and T_f_: final degradation temperature.

**Table 8 polymers-13-02475-t008:** Mechanical properties (Young’s modulus, tensile strength, and elongation at break), draw ratio (DR), and dimensions of neat and functionalized PLA flat tape samples.

Sample	DR	Dimensions (mm) (Width × Thickness)	Tensile Strength at Break (MPa)	Young’s Modulus (GPa)	Elongation at Break (%)
PLA	4.5	2.70 × 0.14	157 ± 10	4.6 ± 0.1	40.2 ± 2.6
O-PLA-A	5.5	2.34 × 0.13	238 ± 5	5.1 ± 0.1	30.9 ± 1.0
O-PLA-B	6.6	2.40 × 0.14	215 ± 10	4.8 ± 0.2	31.9 ± 2.7
P-PLA-A	5.5	2.32 × 0.13	212 ± 15	5.1 ± 0.1	29.4 ± 3.9
P-PLA-B	6.2	2.37 × 0.14	177 ± 8	4.9 ± 0.1	23.4 ± 2.7

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
