# Peer review of "Improved Mechanical, Thermal, and Hydrophobic Properties of PLA Modified with Alkoxysilanes by Reactive Extrusion Process"

_polymers, 2021, doi:10.3390/polym13152475_

Round 1

Reviewer 1 Report

Please find my comments in the attached pdf file.

Thank you.

Reviewer 2 Report

The aim of the article is the evaluation of polylactide modified by reactive extrusion with alkoxysilanes. Even though the paper is interesting and generally well-written, I think that there are some issues which can be improved. Below please find my comments.

  1. I am not a native English speaker, but I do recommend to check and edit the language of the article, especially in the Introduction part.
  2. Reactive extrusion description: How were the extrudates cooled after processing? Or was the extruded PLA directly fed to the single-screw extruder for the tape extrusion? Please specify.
  3. Table 2. Why the draw ratio was not constant for all the samples? I think that it could influence the macromolecular orientation, and, as an effect, the mechanical properties of the materials. I think that in this situation an analysis of the orientation would be helpful.
  4. DSC method description. Why were the samples heated up to 170 degrees during the first run and to 250 degrees during the second run? Was the temperature of 170 high enough to fully erase the thermal history of the material?
  5. Formula (1) - in my opinion this formula should also take into consideration mass of the additive present in each sample.
  6. Lines 215-222 and 311-318: the same paragraph is repeated twice.
  7. Lines 324-326: Determination of Tg by DSC is not very precise. If you want to thoroughly analyse the glass transition of the samples, dynamic mechanical thermal analysis would be a better choice.
  8. Lines 338-339: Frankly, I cannot see this "small shoulder" in the melting peak. Please mark it in the Figure 4.
  9. Lines 357-358. If the alkoxysilanes act as nucleating agents in PLA, they should influence the crystallization temperature. We cannot evaluate it, as the crystallization temperatures are not given in the paper. Do they change at all?
  10. Lines 360-362: In fact, a rheological analysis of the modified materials would be a great fit in this article. It would provide useful information about both the molecular structure and processing properties of the materials. Please consider adding it to the manuscript.
  11. Mechanical properties: As I already said, the changes of the mechanical properties can be connected to the differences of the molecular orientation in the studied materials. I think it should also be evaluated.

Reviewer 3 Report

The work of Torres et al. examined an important topic. The experiments of the authors are well described.  The results were discussed and the conclusions are also sound. I recommend the publication of this work in Polymers after a major revision.

[1] Title, Abstract, Keywords need to be changed according to the main text. At least, the title should include thermal properties as it has been discussed in the main text.

[2] It would be useful to include the surface morphologies of the prepared blends.

[3] The mechanical and hydrophobic results of the prepared materials should be discussed and compared with previous investigations in the literature. The lack of discussion makes it difficult to determine the scientific contribution of the present work.

[4] Relevant literature should be cited. The following examples are highly recommended;

  [a] https://www.sciencedirect.com/science/article/pii/S0079642520300992

  [b] https://www.sciencedirect.com/science/article/pii/S0264127520301374

    [c] https://onlinelibrary.wiley.com/doi/full/10.1002/apj.1802

Round 2

Reviewer 3 Report

The authors have revised the manuscript according to my comments. It can be accepted now.